# Structure–Property Relationship, Glass Transition, and Crystallization Behaviors of Conjugated Polymers

**DOI:** 10.3390/polym15214268

**Published:** 2023-10-30

**Authors:** Tengfei Qu, Guangming Nan, Yan Ouyang, Bahaerguli. Bieketuerxun, Xiuling Yan, Yunpeng Qi, Yi Zhang

**Affiliations:** 1University and College Key Lab of Natural Product Chemistry and Application in Xinjiang, School of Chemistry and Chemical Engineering, Yili Normal University, Yining 835000, China; 2Anhui Key Laboratory of Spin Electron and Nanomaterials of Anhui Higher Education Institutes, School of Chemistry and Chemical Engineering, Suzhou University, Suzhou 234000, China

**Keywords:** conjugated polymers, glass transition, crystallization behaviors, structure–property relationship

## Abstract

Conjugated polymers have gained considerable interest due to their unique structures and promising applications in areas such as optoelectronics, photovoltaics, and flexible electronics. This review focuses on the structure–property relationship, glass transition, and crystallization behaviors of conjugated polymers. Understanding the relationship between the molecular structure of conjugated polymers and their properties is essential for optimizing their performance. The glass transition temperature (*T_g_*) plays a key role in determining the processability and application of conjugated polymers. We discuss the mechanisms underlying the glass transition phenomenon and explore how side-chain interaction affects *T_g_*. The crystallization behavior of conjugated polymers significantly impacts their mechanical and electrical properties. We investigate the nucleation and growth processes, as well as the factors that influence the crystallization process. The development of the three generations of conjugated polymers in controlling the crystalline structure and enhancing polymer ordering is also discussed. This review highlights advanced characterization techniques such as X-ray diffraction, atomic force microscopy, and thermal analysis, which provide insights into molecular ordering and polymer–crystal interfaces. This review provides an insight of the structure–property relationship, glass transition, and crystallization behaviors of conjugated polymers. It serves as a foundation for further research and development of conjugated polymer-based materials with enhanced properties and performance.

## 1. Introduction

In 1977, the discovery of the first conducting polymer, chemically doped polyacetylene appeared. As a result, people’s perspective on electrically active materials underwent a transformation [1]. Since then, conjugated polymers have rapidly developed as an important organic semiconductor, expanding the semiconductor industry dominated by inorganic materials such as silicon in recent years. The great interest in organic semiconductors is partly due to the challenges associated with processing inorganic semiconductor materials. People can fabricate large-area and uniform organic semiconductor films using mild solution-based processes without high vacuum, with an optoelectronic performance comparable to inorganic materials. In addition, the microstructure of these materials can be controlled to a certain extent by selecting the solubility and solution rheology, which have a significant impact on film formation during solution processing. Therefore, the advantages of convenient solution processing ultimately promote the manufacturing of inexpensive, lightweight, flexible, stretchable, and bio-integrated flexible devices [2,3,4].

## 2. The Structure–Property Relationship of the Conjugated Polymers

### 2.1. The Charge Transport in Conjugated Polymers

Compared to inorganic materials, conjugated polymers exhibit different mechanisms in charge transport. Inorganic semiconductors consist of tightly bonded atoms through strong covalent bonds, forming ordered crystal structures. On the other hand, the weak dispersion interactions between molecular components in organic semiconductors make their electrical properties highly susceptible to structural disorder caused by aggregation. Studies have revealed that in organic semiconductors, the degree of structural order plays a crucial role. Charge transport in these materials can occur through various mechanisms, including band-like hopping, localized hopping, or a combination of the two, depending on the level of structural order present [5,6]. Band-like transport is the dominant mode of charge transport in high-crystalline materials. In these materials, the well-organized and orderly structure allows for the efficient movement of charges through extended electronic states, known as bands. This band-like transport facilitates the flow of charges over long distances, resulting in high conductivity. In these cases, the highly ordered arrangement of atoms in the lattice leads to the complete delocalization of band-edge charge carriers, allowing them to travel unimpeded along a path. The charge carriers in a material undergo a net displacement in an electric field. This displacement occurs at a speed that is directly proportional to the strength of the applied electric field. The magnitude of this proportionality is expressed by a property called mobility. The mobility represents the constant of proportionality between the speed of charge carriers and the electric field strength, providing a measure of how easily charges can move through the material under the influence of an electric field. However, in these material systems, mobility decreases with increasing temperature due to the increased concentration of phonons. In contrast, charge transport is heavily influenced by the presence of polymer chains and the nature of the crystalline structure, which exhibits small overlaps of molecular orbitals generated by van der Waals interactions, resulting in an inherently disordered structure that confines charge carriers to localized regions [7].

There are several barriers to charge transport in polymer molecular systems. First, there are static obstacles caused by crystal structure defects or chemical impurities acting as charge traps. Second, electrons are not only influenced by interactions with other electrons but also by the atomic nuclei and lattice vibrations. These vibrations can be considered as the particles’ transmission of energy and momentum within the crystal, and their interaction with electrons affects the movement and energy states of the electrons within the lattice [8,9]. Hence, in organic semiconductors, charge transport does not typically adhere to the band mechanism under normal environmental conditions. This model suggests that potential barriers at grain boundaries within the material intermittently trap and release charge carriers, influencing their movement and overall conductivity [10]. According to this model, there is an assumption that localized energy levels are located close to the delocalized transport band. When charge carriers reach these localized states, they are immediately captured. This distribution of localized energy levels influences the capture and release of charge carriers, affecting their overall mobility and conductivity within the material.

As a result of their specific arrangement of atoms and conjugated backbone, conjugated polymers exhibit different properties and behaviors in terms of how charges move through the material. In addition, the layered structure of polymer chains leads to different charge transport mechanisms at different length scales, from the conformation of individual polymer chains to the filling structure in polymer aggregates (Figure 1). By aligning the unhybridized P_z_ carbon orbitals along a polymer chain, conjugation occurs, leading to the formation of extended π orbitals. These extended π orbitals create a conduit that enables the smooth movement of charges within the same chain, promoting efficient intrachain charge transport within the conjugated polymer structure [11]. Hence, the transport of charges is restricted by the intrachain electron coupling, which is heavily influenced by the conformation of the polymer chains [12]. Polymers with higher main-chain torsional angles or increased flexibility tend to introduce more conformational disorder, leading to a reduction in the effective conjugation length [13,14]. In certain circumstances, charge transport is facilitated by incoherent hopping, wherein charge carriers leap between different localized states. When compared to band-like transport, the mobility of carriers in hopping transport is considerably lower and primarily influenced by thermal activation. Consequently, the mobility of carriers tends to increase as the temperature rises.

### 2.2. The Multi-Scale Aggregated Structure and the Charge Transport of Conjugated Polymers

For typical conjugated polymer systems, interchain charge hopping transport is feasible. Recent studies [15,16] have shown that charge transport along the main chain is also possible in conjugated polymers. In addition to high crystallinity and ordered arrangement, the polymers used in most studies have highly planar main chains and low trap densities. The π orbitals of adjacent chains align and stack together in the microcrystals formed by the polymer chains. This stacking arrangement creates a pathway through which charges can be transported between the chains, enabling interchain charge transport. However, it is important to note that within these ordered regions, the transport of charges is constrained by the process of interchain hopping. Compared to intrachain hopping, interchain hopping is considerably slower, typically occurring at a rate that is several orders of magnitude more sluggish. Enhancing the electronic coupling between polymer chains through polymer synthesis or processing methods can increase the charge carrier mobility within this length scale, allowing the polymers to form larger aggregates or microcrystals with closer π-π stacking distances. Moreover, the presence of an ordered arrangement of polymer chains is advantageous because it reduces the occurrence of interchain charge hopping. Interchain hopping is a slower process that demands higher energy compared to intrachain transport [17]. In cases where the length scale is larger than microcrystals, carrier transport is determined by hopping between microcrystals/aggregates. The band gap in the amorphous regions is often different from those in the crystalline domains, resulting in high energy barriers for charge movement at the interface between ordered and disordered regions [18]. All collectively contribute to the overall structure of thin films made up of organic semiconductors, and it is essential to take all these factors into consideration when assessing the performance of devices that incorporate them, as illustrated in Figure 2. Additionally, according to the findings of Mollinger and Spakowitz, their research indicates that the conjugation length of the polymer should be sufficiently extended to bridge the gap between microcrystals. This bridging effect enables a continuous flow of charges through interconnected chains, facilitating uninterrupted charge transport [19]. In a recent study conducted by Gu and Loo, they highlighted the significance of a critical fraction of interconnected chains (poly(3-hexylthiophene) (P3HT) is 10^−3^) for achieving efficient charge transport in conjugated polymer chains [20,21]. When the fraction of interconnected chains falls below the critical threshold, charge transport is hindered by the connectivity between the grains. On the other hand, when the fraction surpasses the critical threshold, charge transport is primarily influenced by the disorder present within the grains.

The amorphous and crystalline regions of conjugated polymers display distinct electronic properties. The band gap, which originates from the interaction between the π and π* orbitals of the repeating units along the polymer chain, can be altered by conformational changes. Consequently, variations in conformation contribute to modifications in the electronic levels of the polymer. For example, the electronic level of crystalline RR-P3HT (5 eV) is lower than that of the amorphous RRa-P3HT (5.25 eV) [23]. The presence of conformational disorder has a disruptive effect on intermolecular interactions, consequently altering the electronic levels. This disorder, be it structural or electronic, broadens the electronic density of states (DOS) and gives rise to trap states that hinder carrier mobility. The differing preferential orientations of ordered structures result in significant anisotropy in mobility, ultimately affecting carrier transport. During the crystallization process of P3HT in a film, two typical molecular orientations can emerge. The first one is known as the edge-on configuration, which is often observed when P3HT is deposited on surfaces like the amorphous ITO glass substrate. This preference for the edge-on configuration can be attributed to the relatively weak interaction between the P3HT molecule and the substrate [22]. The edge-on configuration is particularly well-suited for devices like OFETs and organic sensors. These devices require efficient charge transport along the in-plane direction of the semiconductor. The presence of π-π stacking, where the semiconductor aligns parallel to the substrate, significantly enhances charge transport in this direction. As a result, the edge-on configuration proves advantageous for enhancing the performance of such devices. However, organic photovoltaic devices and organic light-emitting diodes rely on the face-on configuration to achieve high efficiency. These devices function through vertical charge separation or charge injection processes. Recently, notable progress has been achieved in the advancement of high-performance organic solar cells through the utilization of the face-on orientation of the polymer. This orientation has paved the way for significant improvements in the efficiency and effectiveness of organic solar cell technology. This configuration is significant for optimizing the performance of these optoelectronic devices [24,25,26]. Moreover, it is worth noting that a predominant edge-on packing arrangement is observed in the majority of high-performance semicrystalline p-type polymers [11,27,28].

It is widely recognized that various high-performance polymers exhibit distinct preferred morphologies in the device, encompassing molecular orientation in relation to the substrate, crystallinity, and domain size. For example, regio-regular P3HT has a tendency to form edge-on lamellae in P3HT:PCBM films and possesses a higher crystallinity compared to other donor-conjugated polymers. These characteristics are directly associated with its high photovoltaic performance [29,30]. On the other hand, in several newer high-performance donor polymers like thienothiophene (TT) and benzodithiophene (BDT), as well as BDT- and N-alkylthieno[3,4-c]pyrrole-4,6-dione (TPD)-based co-polymers [31,32], the preferred orientation to the substrate is face-on [33,34]. This packing orientation is deemed more beneficial for hole transport in vertical diode configurations commonly employed in photovoltaic applications [34,35]. The alignment of crystallites, whether oriented as face-on or edge-on relative to the substrate plane, significantly influences the charge transport properties of semi-crystalline conjugated polymer thin films. In the case of RP-3T, the as-cast film primarily exhibits a face-on orientation. Nevertheless, when subjected to annealing temperatures below 194 °C, a remarkable shift towards an almost exclusive edge-on orientation is observed. This transition in orientation has a substantial impact on the charge transport characteristics of the thin film, thereby altering its overall charge transport properties [36].

The complex, multi-layered structure of conjugated polymers is of central importance in charge transport [18]. This process starts with the formation of single polymer chains. Various factors including the spacing between chains in the aggregate structure, the size and layout of the aggregated structure, and its overall interconnectivity, all collectively dictate the charge transport proficiency of conjugated polymers. Typically, the process by which these polymers form highly structured formations through stacking tends to be slower than the time necessary for polymer processing. In other words, the chain structure of conjugated polymers typically cannot rapidly form equilibrium structures during processing (such as spin coating or brush coating), making it challenging to achieve ideal ordering. Ideal molecular ordering encompasses main chain planarization, high crystallinity, transition pathways between crystals, and the arrangement of aggregated structures. Therefore, extensive efforts have been made from molecular design to process innovation to achieve regulated and ordered rearrangement of conjugated polymer structures at different scales. Consequently, researchers have developed relevant control techniques in studying the fundamental processes of polymer crystallization and the formation of ordered aggregated structures.

## 3. The Glass Transition of the Conjugated Polymers

### 3.1. The Glass Transition in Polymers

As a polymer transitions from a molten state to a solid state, there is a corresponding gradual reduction in its specific volume. As the temperature decreases, the molecular segment mobility also decreases. In the molten state, the polymer chains have sufficient mobility to maintain equilibrium. However, as the cooling temperature decreases, the mobility of the segment becomes so low that equilibrium cannot be reached within the measurement time scale (e.g., seconds, depending on measurement conditions). This leads to a deviation from the equilibrium volume and the polymer enters the glassy state, which is characterized by the glass transition temperature (*T_g_*), as shown in Figure 3. *T_a_* represents a temperature lower than *T_g_*, at which physical aging experiments can be conducted, while *T_f_* is a fictive temperature used to describe the thermodynamic state of the polymer chain segments.

With its long chain nature, the glass transition temperature of the polymer shows a dependence on the molecular weight. For linear polymers, the glass transition temperature increases with an increase in the number-average molecular weight (*M_n_*), as described by the empirical Fox–Flory equation [38].
$$T_{g}(M_{n})=T_{g(\infty )}-\frac{K}{M_{n}}$$
where *T_g_*_(∞)_ represents the glass transition temperature for a polymer with an infinitely high molecular weight, while *K* denotes a constant value.

The stiffness of the polymer chains is also an important factor influencing the *T_g_* of polymers. As the length of the polymer chains increases, the chains can become either more rigid or more flexible [39,40,41,42]. There are two factors that affect chain stiffness. One factor is the flexibility of the backbone main chain, which is related to the ease of rotation of the segments. For example, the main chain of poly(dimethylsiloxane) is highly flexible due to its low rotational energy barrier, resulting in a glass transition temperature of −123 °C. Another example is polyethylene, which has a glass transition temperature of −90 °C [40]. The second factor is the influence of side chain groups, which increase chain stiffness due to steric hindrance effects. For instance, replacing hydrogen atoms with methyl groups in polyethylene leads to an increase in *T_g_* from −90 °C to −20 °C, and the larger phenyl groups in polystyrene further increase the glass transition temperature to 100 °C [43]. However, long alkyl side groups increase the flexibility of the polymer chains by introducing more free volume, thereby reducing the *T_g_*. When the number of alkyl carbon atoms exceeds twelve, the *T_g_* begins to increase again due to side chain crystallization, which reduces the mobility of the polymer chains [44]. In addition, some references indicated that side chain crystallization is observed in several atactic polymers when there are approximately 8–12 alkyl carbons per side chain. For instance, the polyethylene-like glass transition temperature of the alkyl side chain through DMA tests in P3HT is approximately −100 °C, in P3OT is around −65 °C, and in P3DDT is about −50 °C [45]. The interpretation of the phenomena is that the alkyl nanodomains are formed by the self-assembled of the alkyl side chains, and the CH_2_ units of the side chain are controlled by the size of the nanodomain confinement. However, the values for *T_g_* obtained from DSC experiments are P3BT (34 °C), P3HT (4 °C), and P3OT (−22 °C) [45]. This phenomenon indicates that different testing methods yield varying glass transition temperatures for the corresponding sample’s microscopic structures (side chains or main chains). On the other hand, it underscores the significance of the measurement method in determining the glass transition temperature of materials.

In semicrystalline polymers, the influence of stereospecificity; crystallinity; and orientation in the absence and presence of crystallinity on the glass transition temperature is an important area of research [46]. The amorphous fraction of polymers can be characterized by three distinct features: orientation, density, and confinement. In the case of semicrystalline polymers, they consist of three main components: the crystalline fraction (CF), the mobile amorphous fraction (MAF), and the rigid amorphous fraction (RAF). The rigid amorphous fraction is considered near the crystalline region, separated from the CF and the MAF. The chain segments in the RAF are constrained by the crystalline regions, while the chain segments in the MAF are unaffected by the crystalline regions. As a result, the *T_g_* of the RAF is higher than that of the MAF [47]. This has been observed in recent studies on conjugated polymers, such as P3HT [48,49].

### 3.2. Experimental Techniques for Measuring the T_g_ in Polymers

Experimental methodologies for determining the *T_g_* encompass dynamic measurement techniques like broadband dielectric spectroscopy and dynamic mechanical analysis (DMA). In DMA experiments, samples in a “dog bone” shape are prepared, and temperature measurements are performed at a specific heating rate to obtain the *T_g_*. Due to the dynamic characteristics of the glass transition, the numerical value of the *T_g_* according to the experimental frequency increases with higher frequencies. These experimental procedures are optimally designed for samples weighing 100 mg or more. Researchers [50] have improved the DMA instrument setup, as shown in Figure 4, by drop−coating the conjugated polymer onto glass fibers. The peak in the tan *δ* response corresponds to the glass transition and sub–*T_g_* (sub–glass transition) transition. The results obtained from this study enhance our knowledge and enable improved management of the microstructure and thermal behavior of blends containing conjugated polymers.

Thermodynamic measurement techniques for measuring the glass transition temperature include volume dilatometry and differential scanning calorimetry (DSC). DSC is widely used for measuring the *T_g_* of polymers [51]. Generally, when measuring the glass transition temperature, the process involves two measurement loops. During the initial measurement loop, the sample is heated to a molten state to eliminate any thermal history, such as physical aging in amorphous polymers, and bring it to a molten state. For semicrystalline polymers, it is necessary to heat the sample into a molten state. Then, it is cooled at a defined cooling rate (e.g., 10–20 °C/min) to a low temperature (below *T_g_*) and then reheated at the same rate to a high temperature, from which the *T_g_* is determined in the second heating scan. Measuring the *T_g_* is commonly performed by determining the heat capacity (Δ*C_p_*) in the heating scan. This method is widely accepted in practice.

### 3.3. The Fictive Temperature in Glass Transition

*T_f_* is a fictive temperature used to describe the thermodynamic state of the materials. The *T_g_* is typically determined through cooling experiments exclusively [52,53]. In fact, heating scans in the DSC experiments can provide a fictive temperature *T_f_*, proposed by Tool [54], initially used to study the thermodynamic state of glassy material and can be obtained using the Moynihan method [55]. The sample without physically aging is cooled to the glassy state and immediately heated at the same rate, resulting in a heating scan curve that yields a limit fictive temperature *T_f_*, which is equivalent to the *T_g_* obtained during the cooling process. Even in the fastest quenching scenario, where the fictive temperature describes the energy probability distribution of the glass, the glassy state, and the liquid state are macroscopically different states with different entropy values. Tool’s research [54] indicates that comprehending the *T_f_* in relation to the physical temperature is crucial for obtaining a comprehensive understanding of the glassy state. This insight provides a more complete description of the properties and behavior of glasses. If the sample is quenched instantaneously from the molten state, directly forming a glass from the equilibrium liquid phase, the hypothetical temperature represents the temperature of the melt before rapid cooling. During the quenching process, the temperature rapidly drops within the range of the glass transition, leading to the solidification of the liquid. After quenching, the dynamics of the sample are controlled by the experimental temperature T.

According to reference [56], there are at least three definitions for the glassy state structure of materials: on a microscopic level, mapping the structure of the non-equilibrium disordered state to a corresponding structure found in an equilibrium liquid phase; on a macroscopic level, representing the properties of the glassy equilibrium state using the values of equivalent configurational properties; from a dynamic perspective, the fictive temperature is utilized to characterize different physical recovery behaviors of the structure of the glassy state. The interpretation mentioned above enables the description of simplified statistical mechanical models for materials in the non-equilibrium glassy state. This is achieved by utilizing linear combinations of properties derived from the equilibrium liquid state. Such an approach makes it feasible to capture essential characteristics of the non-equilibrium glassy state using properties observed in the equilibrium liquid state. The *T_f_* is used to describe the rapid cooling of a sample through the glass transition region. By tracking the change in fictive temperature, one can gain insights into the structural and dynamic properties of the glassy state. The corresponding enthalpy and molar volume can describe the properties of the sample. However, the concept of the distribution of *T_f_* hardly accurately obtains the fluctuations in physical and chemical parameters like enthalpy and volume. Sometimes, enthalpy distribution states are also used to describe the glassy state, providing a true statistical thermodynamic interpretation of materials in the glassy state. According to Narayanaswamy, the *T_f_* of glass material is influenced by its crystallization and stress during production due to storage conditions or environmental influences prior to calorimetry. It is also affected by the changes with variations in temperature specified, such as during the process of calorimetry around the *T_g_* [57]. This can help to characterize the structural state of the glass through its fictive temperature. It is believed to reflect the temperature at which the glass needs to be rapidly cooled in order to achieve its particular structural state.

### 3.4. The Measurements of Tgs in Conjugated Polymers

The study of glass transition in polymer film samples can be conducted using different types of DSC depending on the size parameters of the film, such as film thickness. Conventional DSC techniques can achieve cooling speeds that are under 50 K per minute at their maximum. It is typically applied to measure bulk samples and is also an important tool in the measurement of polymer ultra-thin films at the nanoscale. However, sample preparation can be a time-consuming process, as it involves the precise stacking of hundreds of layers of ultra-thin films to meet the necessary mass requirements [58,59]. With the development of nano-calorimetry techniques [60], including alternating current (AC) chip calorimetry and fast scanning calorimetry (FSC) [61,62], dynamic measurements of *T_g_* are possible. Sample masses as low as nanograms can be used, allowing for the measurement of *T_g_*s in ultra-thin films with thicknesses below 100 nm [63,64]. The AC chip calorimetry technique was a method to observe the glass transition of Diketopyrrolopyrrole (DPP)-based polymers, which had film thicknesses of 110 nm, 72 nm, and 33 nm, respectively [64]. Meanwhile, there is growing interest in polymer ultra-thin films [65,66,67].

While there are various testing methods for determining the *T_g_* of conventional polymers, there is a notable scarcity of information regarding the *T_g_* of conjugated polymers in the existing literature. This knowledge gap hampers an overall comprehension of the mechanical properties and morphological stability, particularly concerning the applications of organic electronic devices [68,69]. This is mainly because obtaining the *T_g_* of conjugated polymers through traditional testing methods such as DSC is challenging. The rigid and twisted backbone structure of conjugated polymers poses a challenge for conventional DSC in accurately capturing weak backbone relaxation signals. To overcome this, reliable signals, particularly for conjugated polymers with donor groups and acceptor groups, can be measured using FSC or AC chip approaches. Secondly, batch synthesis experiments are typically capable of producing only a restricted quantity of samples, usually around 20–30 mg. The rheological or dynamic mechanical analysis tests demand a larger amount of material, exceeding 100 mg. Furthermore, unlike traditional amorphous polymers, conjugated polymers incorporate different structural units into their backbone and side chains through structural engineering. Therefore, it is extremely difficult to establish a universal model for predicting the *T_g_* of them. When discussing the glass transition of conjugated polymers, it is necessary to first grasp the relaxation behavior of their different chemical structures and groups. For instance, polythiophene-based polymers, the *T_g_*s of backbones can vary above 80 °C [48,70]. Therefore, it is important to divide conjugated polymers with different chemical structures to better comprehend the effect of different structural units on the device performance.

#### 3.4.1. The Effect of the Lengths and Branches of the Side Groups on *T_g_* of Conjugated Polymers

In the early 1990s, the synthesis of poly(3-hexylthiophene) (P3HT) was achieved [71]. Since its inception, it has consistently been considered the most extensively studied sample in the literature. In contrast to the conjugated main chain that determines the photoelectric function of conjugated polymers, the initial introduction of flexible solubilizing side chains was primarily aimed at facilitating synthesis and processing. In the past few years, there has been a growing awareness of the significance of side chains in forming nanocrystalline domains and influencing the mechanical properties of conjugated polymers. This has sparked a surge in research exploring the functional aspects of side chains. As a result, there is a noticeable increase in research interest in understanding the role of side chains in this context [72]. For example, poly(alkylthiophene)s have been used as ideal models to investigate the impact of different side chains on *T_g_*. Generally, the introduction of longer side chains increases the free volume and configurational entropy, thereby reducing the *T_g_* of the main chain. Among them, the *T_g_* of poly(3-butylthiophene) is 45 °C, poly(3-hexylthiophene) is 12 °C, and poly(3-octylthiophene) is −13 °C [45]. However, as the side chains further lengthen, *T_g_* increases due to side chain crystallization, similar to poly(4-alkylstyrene) [44]. The backbone *T_g_* of P3ATs depends on the regularity of the polymer chain. The polythiophene with side chains, and the insertion of the center-symmetric repeat unit thiophene-benzothiadiazole as a main chain structure to enhance the molecular order and device performance may also influence *T_g_*. Taking poly(2,5-bis(3-alkylthiophen-2-yl)thieno[3,2-b]thiophene) (PBTTT) as an example [73], the additional free volume generated by the thiophene-benzothiadiazole core without substituted side chains allows for filling-in of the side chains from close molecules [74]. This side chain interdigitation promotes the fill order between the main chains of PBTTT, which is an important part of its glass transition process [75,76].

The *T_g_* of the polythiophene polymers decreases as the side chains, especially their lengths, appear. This helps to improve the mechanical behaviors of flexible devices made from conjugated polymers. When alkyl side chains are branched in the conjugated polymer, the influence on the glass transition of the main chain is mainly considered of their steric hindrance and the role of different length side chains on the movement of the main chain segments. For example, the *T_g_* of poly(3-(4′-methylpentyl)thiophene) (P3(4MP)T) (36 °C) is lower than that of poly(3-butylthiophene) (P3BT) (60 °C) but higher than that of P3HT (16.3 °C). This is because the steric hindrance of the alkyl side groups in P3(4MP)T is greater than the one of the hexyl alkyl side chains in P3BT. The carbon tetraalkyl side chain in P3BT is too short to effectively improve the movement of the main chain segments, resulting in a higher glass transition temperature compared to P3(4MP)T. Based on the effect of the steric hindrance, it can be observed their glass transition temperatures show a decreasing trend from high to low, poly(3-(2′-ethylbutyl)thiophene) P3(2EB)T > poly(3-(3′-methylpentyl)thiophene) P3(3MP)T > P3(4MP)T > poly(3-(2′-methylpentyl)thiophene) P3(2MP)T, as shown in Table 1 [45,77,78,79,80].

#### 3.4.2. The Influence of the Rigidity of the Main Chains on *T_g_* of Conjugated Polymers

Diketopyrrolopyrrole (DPP) based polymers are important representatives of donor-acceptor (D-A) conjugated polymers. Different main chain structures of DPP-based polymers are ideal objects for studying their glass transition behavior, which is important for molecular designing and improving their performance in electronic applications. The various main chain structures in these polymers have their own influences on the glass transition behavior and overall properties. One common main chain structure in DPP-based polymers involves alternating conjugated units, such as thiophene or benzene rings, with DPP units. This alternating structure allows for efficient π-electron moving along the polymer chain, which helps to enhance carrier transport characteristics. In terms of glass transition, the presence of conjugated units often increases intermolecular interactions and molecular order within the material, leading to higher glass transition temperatures. This promotes a more rigid, glassy state at 25 °C.

Measuring the *T_g_* of DPP-based polymers is challenging due to their complex main chain structures. However, AC chip measurement provides a possibility for this research. The different rigidity of thiophene rings, fused thiophene rings, and fused triple thiophene rings in the main chains of P(DPPT), P(DPPTT), and P(DPPTTT), respectively, as well as the differences in volume space they provide for the side chains, gradually increase the steric hindrance, resulting in an increment in the *T_g_* of the main chain amorphous fraction in different conjugated polymers (*T_g_*_,m_: P(DPPT) < P(DPPTT) < P(DPPTTT)). At the same time, the volume space they provide for the side chain segments and main chain segment movement gradually increases, reducing the spatial resistance to the movement of the main chain rigid amorphous fraction (RAF), inducing a reduction in the *T_g_* of the main chain RAF in these polymers (*T_g_*_,r_: P(DPPT) < P(DPPTT) < P(DPPTTT)), as shown in Table 1 [65,81]. In addition, researchers have found through DMA studies that the *T_g_* increases as the size of the fused ring structure enlarges, with *T_g_* values of −3.96 °C (DPPT), 2.76 °C (DPPTT), and 4.11 °C (DPPTTT) [81]. Therefore, these two different research methods (AC chip and DMA) provide detailed study content to deepen the understanding of the glass transition process in DPP-based polymers.

The alkyl-substituted indacenodithiophene (alkyl-IDT) family of conjugated polymers is characterized by a low elastic modulus, high charge transport capacity, and low crystallinity. Among PIDTBP (−8.1 °C), PIDTTPD (−0.6 °C), and PIDTBTD (17.6 °C), PIDTBPD exhibits the lowest *T_g_*, displayed in Table 1 [82]. This is attributed to the stronger intermolecular interactions in PIDTTPD and PIDTBTD compared to PIDTBPD, as discovered through UV–visible spectroscopy. This characteristic is on account of the long alkyl side chains and the distortion of the PIDTBPD polymer backbone, resulting in the predominance of an amorphous state.

Additionally, to gain a deeper understanding of the relationship between the molecular structure and the charge transport capabilities of these promising near-amorphous D-A polymers. Researchers employed a combination of photothermal deflection spectroscopy (PDS) measurements and mechanical property characterization like DMA. This comprehensive approach allowed for a detailed analysis of the micro-structural features that contribute to the favorable charge transport properties of these polymers. In order to examine the relationship between the degree of interchain interactions, probed through sub-bandgap absorption transitions in PDS spectra, and the glass transition temperature, DMA measurements were performed on this polymer series. The observed correlation can be anticipated, as the cooperative movements of the main chains, taking place above the glass transition, necessitate adequate thermal energy to disrupt these interchain contacts. This correlation highlights the importance of thermal energy in influencing the interchain interactions and the overall glass transition behavior of these polymers. Although the four polymers share the same C16 alkyl side chains, significant differences were observed in terms of glass transition and melting processes, even though the secondary relaxation occurs at −20 °C, resulting in an increment in *T_g_* being observed, C16 IDT-BT < C16 TBIDT-BT < C16 NDT-BT < C16 TIF-BT, with C16 TIF-BT exhibiting the highest *T_g_* of 141 °C, as shown in Table 1 [13]. By integrating PDS, mechanical property characterization, and DMA findings, researchers aimed to unravel the underlying mechanisms supporting the outstanding charge transport performance of these near-amorphous D-A polymers.

Significant advancements have been made in understanding and utilizing electron-conducting π-conjugated polymers containing naphthalene diimide (NDI) blocks. These polymers demonstrate enhanced n-doping characteristics attributed to the substantial electron deficiency within the polymer main chain and the existence of positively charged side groups that effectively stabilize the negative charge on the n-doped main chain. In poly(azomethinenaphthaleneimide)s (poly(AZ-NDI)), there is little difference in the glass transition temperature between poly(AZ-NDI)-3 and poly(AZ-NDI)-4, both of which occur around 124 °C, as shown in Table 1 [83]. This can be attributed to the naphthalene diimide structure present in their main chains. Due to the limited literature available on the topic of glass transition of electron-conducting π-conjugated polymers with naphthalene diimide (NDI) blocks and tetrathienoacene-based polymers, further research is still required to fully understand their glass transition.

The *T_g_* is crucial for the practical application of conjugated polymers [84]. Due to significant differences in the molecular structure between the aromatic structure and alkyl side groups of conjugated polymers, the influence of molecular structure on the glass transition temperature of different conjugated polymers requires further investigation. For designing flexible, stretchable, or flexible electronic devices, the glass transition temperature serves as a temperature indicator for the brittle–ductile transition of conjugated polymers. The *T_g_* of the samples affects the movement of molecular chain segments and mechanical properties. Below the *T_g_*, their storage modulus is approximately 1 GPa [85]. At higher temperatures, the storage modulus decreases by approximately ten orders of magnitude, with the storage modulus of semicrystalline polymers at 10 MPa [86]. Therefore, one of the fundamental design factors for flexible electronic devices is guaranteeing the desired performance of the material under normal operating conditions [87,88]. Additionally, conjugated polymers used in various electronic devices [89,90] undergo solution processing, leading to non-equilibrium molecular chains of conjugated polymers that tend to undergo structural evolution or physical aging over time. The dynamics of microstructural evolution during processing and usage are influenced by molecular chain motion, which is also affected by the glass transition temperature of the conjugated polymers themselves. Annealing above *T_g_* can result in the cold crystallization of conjugated polymers [91,92], enhanced liquid crystallinity or thermally induced metastable phase ordering [73], and significant phase separation with the use of other additives or polymer blends [93,94,95]. Therefore, the *T_g_* is an important sign of the embrittlement of conjugated polymers and a critical temperature influencing the morphological evolution in practical engineering, thus affecting the physical performance of conjugated polymer flexible devices.

Researchers have found a strong correlation between the dynamic mechanical process and the side groups of conjugated polymers through dynamic mechanical analysis experiments [72]. For example, in dynamic mechanical analysis studies [96,97,98], the different chemical structures of poly[N-9′-hexadecanyl-2,7-carbazole-5,5-(4′,7′-di-2-thienyl-2′,1′,3′-benzothiadiazole)] (PBnDT−FTAZ) correspond to changes in the relevant modulus, which can be divided into the main chain’s α relaxation temperature (*T*_α_), the main chain’s β relaxation temperature (*T*_β_), and the side chain’s γ relaxation temperature (*T*_γ_), as shown in Figure 5 [96]. The *T*_γ_ in conjugated polymers not only significantly affects the motion of the side chains but also has implications for the deformation behaviors of the polymers and the electronic function of other stretchable devices [98,99,100].

Researchers have discovered that in dynamic mechanical experiments, the modulus of conjugated polymers decreases significantly near the characteristic temperature of the side chains caused by an increment in their density. The characteristics of the side chains make them more prone to sliding [98], as shown in Figure 6 [96]. The presence of side chains allows conjugated polymers to exhibit a certain level of stretchability in environments below the *T_g_* of the disordered side chains. Below the sub−glass transition temperature (Sub-*T_g_*) of the conjugated polymer, deformation is achieved through a slipping system along the densely packed direction of the side chains, resulting in energy dissipation [101]. Additionally, the strength of mechanical relaxation in conjugated polymers is also influenced by the side chains. The mechanical relaxation behavior is significantly influenced by their density and inherent in the chemical structure. For example, the *T*_γ_ in non-thienyl-substituted benzotriazole (noT–FTAZ), at 241 K, is lower than that of PBnDT–FTAZ (249 K). This is due to the greater steric space occupied by the thiophene ring in the side chains of PBnDT−FTAZ, which reduces the available free volume and increases the steric hindrance to side chain motion, as shown in Figure 6 [96].

Due to the temperature influence on the orientation domain size of conjugated polymers along the backbone and side groups directions, the mechanical properties undergo changes in terms of plasticity [102]. For regioregular P3HT and PBnDT−FTAZ, frozen, regioregular P3HT exhibits tubular cracks and toughness changes at 243 K and 163 K, mainly caused by the relaxation behavior of its side groups. PBnDT−FTAZ’s fracture behavior and features at 298 K and 243 K also originated from the motion of the side chains, as shown in Figure 7 [98]. Moreover, there has been a proposition of a link between the glass transition and localized molecular motion [103].

In the study of conjugated polymers, the glass transition temperature appears to increase with increasing chain stiffness and larger side groups and decrease with increasing side chain length [45,104]. The study of this phenomenon is crucial for analyzing the ductile–brittle transition phenomenon of materials and optimizing the long-term service applications of various forms of materials (coatings, shape tools, special components). In summary, deeper study is required to investigate the polymer relaxation behavior, especially the low-temperature relaxation behavior of conjugated polymers, near the *T_g_* of the various chemical structures of the main chains and so on.

## 4. The Crystallization Behaviors of the Conjugated Polymers

### 4.1. The Multi-Scale Morphology and Crystallization of the Conjugated Polymers

To achieve precise control over the multi-scale morphology of conjugated polymers, it is crucial to have a deep comprehension of their nucleation and growth, as well as the subsequent formation of multi-scale ordered structures. However, based on their complicated crystallization behavior, a widely recognized theory has not yet emerged. The theoretical framework contains the nucleation growth of the crystal and the nucleation growth of the conjugated polymer. It has been carried out on the fundamental principles of nucleation growth and the nucleation growth of the crystal in conjugated polymers. The different crystallization behaviors of flexible and/or rigid molecular chains are a very important context. By manipulating the different crystallization stages and conducting electrical performance tests, researchers have obtained relevant information regarding the aggregation structure, packaging, and device properties. For example, an important advance has been made in studying the crystalline morphology of P3HT.

Regioregular P3HT is a common thiophene-conjugated polymer, in which each repeat unit consists of the same isomer (3-hexylthiophene) with a high regularity of up to 98%. Large-scale synthesis of P3HT has been achieved at room temperature. Research has shown that the molecular structure of P3HT thin films plays an important role in their optical, carrier transport, and electrochemical properties [104,105]. Changing solvents, film formation techniques, and annealing processes are the main methods for controlling the structure of films [106]. Annealing processes have been widely applied to pure P3HT films, as well as films containing fullerene blends and their impact on solar cells, at 150 °C isothermal no more than 30 min [107].

The crystallization process of P3HT is primarily controlled by strong π-π interactions between each perpendicular direction on the thiophene rings of the main chain. This driving force promotes the isotropic growth of stacked aggregated polymer chains, as shown in Figure 8 [108]. The π-π stacking occurs in the [010] direction, with the molecular chains lying in the plane. Thin, folded crystalline regions and end terminations separated by amorphous regions provide effective in-plane transport pathways along the π-π stacking direction, enhancing the electrical performance of organic semiconductor devices. The rate of charge transfer is determined by the slowest process, which encompasses the transfer of charges from one ordered domain to another, including the crossing of crystalline-amorphous region interfaces and ordered domain boundaries [109]. The rate of carrier transport is governed by the slowest process, which involves the transfer of charges between ordered domains. This encompasses the crossing of crystalline-amorphous interfaces as well as domain boundaries. Ultimately, it is the efficiency of these charge transfer events that determines the overall rate of charge transfer. Researchers aim to improve the electrical properties of conjugated polymers by manipulating their crystalline structure. One approach is to continuously prepare conjugated polymer thin films through a drying process at reduced temperatures or increased solution concentrations. Another approach involves heating the conjugated polymer to a molten state and recrystallizing preformed films under specific isothermal conditions [110]. By observing the crystallization process through film preparation, the number of nucleations varied and decreased, and uniform geometric structures of films can be attained, with sizes extending to dimensions of macroscopic carrier transport, typically ranging in the tens of micrometers [111]. Importantly, this method allows for the control of nucleation density. These results confirm the expected nucleation and growth behavior during P3HT film growth, making it an ideal case for studying the relationship between conjugated polymer charge transport and aggregation structures.

It was believed that polymer chains formed lamellae through chain folding crystallization in 1938 [112], which then developed into single crystals or spherulites (Figure 9d). In the 1960s, Keller and colleagues [113] provided evidence for chain folding in polyethylene crystals, challenging the previously accepted tassal−microbeam model proposed by Bryant. Through further research [114], numerous theories have been formulated to elucidate the process of polymer crystallization. It is a dynamic process, and the polymer crystal structures often exist in metastable states. Although there may be variations in the specific mechanisms of chain folding and subsequent polymer crystallization among different theories, they are based on a common foundation called the Hoffman–Lauritzen (HL) theory [115].

Many theories developed subsequently are built upon the foundation of the Hoffman–Lauritzen theory to explain more complex crystallization processes under different conditions or when considering additional factors in polymer crystallization [120,121]. Wenbing Hu’s “Principles of Polymer Crystallization” provides a systematic introduction to the thermodynamics, kinetics, and morphology of polymer crystallization from a microscopic molecular perspective [122].

Conjugated polymers have unique structural units and microcrystalline morphologies are different from polyethylene. The key characteristic of conjugated polymers is the presence of alternating carbon-carbon double bonds in the main chain. The molecular structure described above is readily apparent in examples of PA and PPP [116,118,123], such as polyacetylene with a linear structure or poly(p-phenylene) with an aromatic structure (Figure 9a). The presence of conjugation in the polymer strengthens the π-π stacking, resulting in increased rigidity. In fact, early research categorized them as “rigid-bar” polymers due to this characteristic [123,124,125,126].

The characteristic of the second generation of optoelectronic polymers is the addition of side chains improves solubility and reduces the difficulty of device processing like P3HT (Figure 9b). Conjugated polymer side-chain research is currently a highly active research field. Its significance lies not only in upgrading the dissolution process but also in altering the formation of microstructure and special properties of the polymer. In fact, P3HT stands out as one of the few polymers that is currently obtainable in quantities exceeding 10 kg [127]. This makes it one of the limited viable options for commercial OPV applications. Its successful utilization has already been extensively illustrated in the preparation of solar cells [128].

Currently, third-generation conjugated polymers showcase more intricate molecular structures. Generally, they are composed of alternating conjugated units that function as both electron donors and acceptors (as shown in Figure 9c). As a result, they possess improved electronic properties, attributed to the increased electron delocalization along the main chain. The molecular segments of typical D-A molecules are characterized by high rigidity [129]. Rigidification may be further enhanced by designing fully fused ladder-type polymers [130]. Hence, conjugated polymers have complicated molecular chain ordering behavior due to their unique molecular design. It is still under exploration whether the polymer crystallization theories established are suitable for conjugated polymers. In P3HT, chain folding does appear, and under given conditions, spherulites form, although fibrous agglomerates are typically observed [119]. The typical morphology of P3HT, as shown in Figure 9e [119], is semi-crystalline. As the third-generation conjugated polymer P(DPPTTT), it exhibits a two-step crystallization phenomenon, displayed in Figure 9f [67].

Currently, there is no mature foundational theory regarding the crystallization of conjugated polymers. Most of the research on conjugated polymers is focused on processing methods that control the molecular order and on studying changes in the micro-scale morphology. Therefore, there have been studies discussing how polymer rigidity and side chains influence polymer crystallization or molecular ordering [129]. Although conjugated polymers are considered to be rigid, apart from some commonly studied examples like P3HT, there is limited literature on the actual measurements of the persistence length (a physical quantity measuring the stiffness of polymer chains) of conjugated polymers [126]. Furthermore, semiconducting polymers possess a higher degree of rigidity, but their stiffness varies. For example, in comparison to D-A conjugated polymers, conjugated homopolymers may exhibit greater flexibility. As a reference, the persistence length of polyethylene is around 0.7 nanometers, while the persistence length of P3HT is approximately 2.8 nanometers [129]. The persistence length of a polymer can be obtained by X-ray small-angle scattering and X-ray wide-angle scattering. Additionally, it can also be analyzed using some methods of computational chemistry. PA and PPP exhibit weak solubility, which makes reports on persistence length relatively scarce, although there are occasional reports of values around 10 nanometers or a few tens of nanometers [131]. The persistence length of second-generation semiconducting polymers typically falls within the range of 2–10 nanometers. In contrast, the persistence length of D-A conjugated polymers is several times that. The microstructure of the folding of chains in P3HT has been confirmed [132]. However, this phenomenon does not appear in D-A conjugated polymers, which may be attributed to their higher rigidity. They form aggregates, along the conjugated main chains aligning along the fiber axis [132].

Miura et al. [133] conducted molecular simulations to study the effect of rigid chains on polymer melt crystallization. Figure 10a illustrates the crystallization process with semi-rigid chains. They found that an increased number of rigid chains improved the ordering of crystalline domains and reduced the crystallization induction time as they could simultaneously elongate and align in a parallel direction. More recently, the crystallization of semiflexible polymers with different volume fractions in melts (φ = 1) and solutions (φ < 1) has been performed by using dissipative particle dynamics, as shown in Figure 10b [134]. Over an extended period of crystallization time, the crystallization process in semiflexible polymers progresses to later stages such as lamellar thickening. This leads to an increase in both the degree of crystallinity and the average size of crystallites, while the number of crystallites decreases over longer time scales. In their work, Yokota and Kawakatsu [135] provided further insights into the impact of chain rigidity on the configurational entropy of polymer chains during both chain folding and chain crystallization processes. For single-chain crystallization, they found that increased rigidity raised the activation barrier for critical nucleus formation. Moreover, compared to single chains, multiple chains were more likely to form critical crystalline nuclei as rigid chains could come together without undergoing extensive chain folding. These studies suggest that rigid chains can promote nucleation, suppress chain folding, and facilitate intermolecular interactions, which also appeared in the D-A conjugated polymers [135]. Figure 10c presents the competition mechanism for orderly arrangement between sidechains and main chains of the conjugated polymer PBnDT−FTAZ. In the spin-coated sample, the sidechains are in a crystalline state currently, and the main chain becomes disordered due to torsion, which obstructs the main chain π-stacking (Figure 10(c-1,c-2)). When the sidechains melt, they are in a disordered state, and the main chain segments can move and adjust their conformation to effectively carry out π-stacking (Figure 10(c-3,c-4)) [136].

### 4.2. The Effect of Side Chains on the Crystallization of Conjugated Polymers

Gaining insights into the impact of side chains on the crystallization of conjugated polymers is an important research topic. Among the alkyl, electron-donating, electron-withdrawing, and conjugated side chains, alkyl side chains are the most utilized in conjugated polymers [76,84]. Originally, the design of side-chain structures aimed to mitigate aggregation and improve the process of the device’s preparation. However, it has since evolved to encompass a wider array of functionalities through more intricate designs. Even slight variations in side chain length or position, seemingly minor in nature, can exert a substantial effect on the crystallization process, the final microstructure, and the properties of conjugated polymers [84]. Carpenter Ade et al. [136] studied the competition between concurrent side chain and backbone crystallization in conjugated polymers, during thin film casting processes. Their findings revealed that side chains had the capability to crystallize into highly coherent layers with coherence lengths exceeding 70 nm. However, this came at the cost of forming torsional disordered layers in the main chain (as displayed in Figure 10c). Panzer and Köhler [137] revealed that these polymers displayed a similar behavior of crystallization when subjected to cooling. Disordered line aggregates expanded due to the main chain twist motion, collapsed into the ordered aggregation by an ordered-to-disordered transition, and eventually underwent side chain crystallization while in the aggregated state.

However, further research has demonstrated that side chains possess more intricate influences on the crystallization process of different conjugated polymers, especially when it comes to the formation of films. For example, Kline and DeLongchamp [100] found that interdigitated side chains in polythiophene films were actually a prerequisite for achieving three-dimensional ordering; without interdigitation, it could not be achieved. The ordered side chains melted into disordered layers, resulting in the crystallization of the ordered main chains (Figure 10c). In the process of designing conjugated polymers, researchers have successfully synthesized poly(2,5-bis(3-alkylthiophen-2-yl)thieno[3,2-b]thiophene) (PBTTT). PBTTT exhibits an optimal density of side chains, ensuring proper interdigitation, and allowing for planarized main chain conformation due to the presence of thieno[3,2-b]thiophene units that facilitate efficient π-π stacking [73,138,139,140].

Poly(2,5-bis(3-tetradecylthiophen-2-yl)thieno[3,2-b]thiophene) (PBTTT-C14) is an important and promising organic semiconductor material. Wang et al. [141] observed three different aggregated states of PBTTT-C14 through thermal annealing experiments at different annealing temperatures, the melting of regularly packed side chains (57 °C), the thermotropic mesophase (133 °C), and the melting of the main chain crystal (223 °C) as shown in Figure 11. Zhu et al. [142] investigated the thermoelectric properties of PBTT-C14 doped films, a conjugated polymer derived from polythiophene, with 2,5-bis(3-tetradecylthiophen-2-yl)thieno[3,2-b]thiophene units. They doped the annealed and non-annealed PBTT-C14 films with ammonium nitrite phosphate. Isothermal treatment increased the number of ordered structures within the films, and this ordered structure was still present during subsequent doping. The presence of these ordered structures contributed to the enhancement of the electrical performance (electronic power index) of the conjugated polymer PBTTT thin films. 

In the literature [143], first-principles calculations were used to simulate the crystal structure of PBTTT-C14. The study revealed that a crystal structure’s bandgap can be adjusted by incorporating side chains with diverse chemical structures. The side chain conformations have little direct impact on the maximum valence band and minimum conduction band of the conjugated polymer, but they do affect the bandgap by altering the π-π stacking distance. Stefania Moro et al., through molecular simulation software, analyzed PBTTT-C14 and successfully fit the model data with experimental structural data of the sawtooth-shaped thiophene and thieno[3,2-b]thiophene main chains, as well as the tetradecyl side chain structures on the thiophene ring [140]. The periodicity along the main chain (13.8 ± 0.07 Å) coincided with data reported in the literature, and the adjacent main chains exhibited highly interdigitated side chain structures [144].

Conjugated polymers exhibit multi-scale morphologies, including intra-molecular conformation, inter-molecular stacking, mesoscale domain size, orientation, connectivity, macroscopic alignment, and (quasi)crystallinity. Isothermal annealing and solvent processing profoundly influence molecular conformation, providing opportunities for manipulating the multi-scale morphologies and carrier transport in semiconducting polymers. The process selection entails utilizing base materials with suitable surface energy to reduce nucleation barriers and promote the orientation of films. This strategic selection of substrates plays a vital role in optimizing the fabrication process. The dynamic nature of the substrate facilitates cooperative motion between multiple bonds and the rearrangement of ordered structures in polymers, accelerating the crystallization process to better match the timescale of the growth of ordered structures in the printing process of conjugated polymers. By significantly accelerating the crystallization process through the aforementioned strategies, new pathways for controlling the crystallization of conjugated polymers can be established. This approach significantly enhances the crystallinity and orientation of conjugated polymers, while controlling various conformational features through surface modification remains challenging. The main key lies in promoting the planarization of the main chain segments, altering the assembly pathways or the degree of aggregation, and aligning polymer films to enhance the electronic property. The directional aggregation, the segment motion, and the crystallization behavior of the main chain in conjugated polymers through annealing or solution processing still require further exploration. Overall, how various micro-morphology control methods influence the transition of the aggregated state of conjugated polymers needs to be further elucidated.

## 5. Conclusions and Outlook

This short review has presented an overview of the research progress concerning the aggregation state structural changes in the charge carrier transport mechanism, glass transition, and crystallization behavior of conjugated polymers. The findings of researchers contribute to a deeper understanding of these key aspects and pave the way for further advancements in the field of conjugated polymers. The iteration and development of conjugated polymers provide a wider range of material choices for the fabrication and application of flexible electronic devices. Therefore, further in-depth research is needed on the aggregation state structural transitions of conjugated polymers with different main chain and side chain structures.

### 5.1. Thermal Behavior Studies of Other Conjugated Polymers

The design and synthesis of conjugated polymers have significantly enriched the source of materials for flexible electronic device fabrication. These conjugated polymers possess a diverse range of main chain and side chain structures. For example, conductive homopolymers include poly(3-bromomethylthiophene-2,5-diyl) and poly([3-thienylpropionic acid]-2,5-diyl). Conductive copolymers include poly([3-thienylpropionic acid]-2,5-diyl) and fluorinated polybenzothiadiazole-thiophene alternating copolymers. Conductive block copolymers include poly(3-hexylthiophene-2,5-diyl)-poly(caprolactone) and poly(3-hexylthiophene-2,5-diyl)-poly(1,4-isoprene), among others. From the preparation and application of flexible electronic devices, it is crucial to study the microstructural changes of these new conjugated polymers in aggregation, as well as the physical and electronic properties.

### 5.2. Glass Transition Studies of Conjugated Polymers

The investigation of the segmental motion mechanisms in the thermal behavior of these new conjugated polymer side chains and main chains pose significant challenges. Combining dynamic mechanical analysis literature, analyzing both dynamic mechanical relaxation and thermal relaxation can provide a deep recognition of the chain segment motion in conjugated polymers. This, in turn, leads to a deeper understanding of the ancient and elusive research on the glass transition of polymers.

### 5.3. Crystallization Behavior Studies of Conjugated Polymers

The microstructure of conjugated polymers exhibits complex and diverse characteristics, which cannot be fully captured solely through static material structure analysis techniques. During service, the temperature, and time-dependent effects, along with the inherent composition, lead to molecular chain segment relaxation or aggregation state evolution, rapid segmental motion, or the emergence of temperature-sensitive metastable structures in conjugated polymers. Therefore, the thermal analysis of conjugated polymers combined with spectroscopic, phase, and structural analysis techniques is significant for studying the aggregation structure, particularly the crystallization behavior (both intrinsic and doped) of conjugated polymers.

The changes in polymer aggregation state structure and crystallization behavior have always been an intriguing and challenging research topic. There are significant differences in the aggregation state structural changes of conjugated polymers under different solvents and temperatures. Hence, the investigation of the glass transition and crystallization behavior of conjugated polymers become more and more significant in the field of flexible electronics and smart skins.

## Figures and Tables

**Figure 1 polymers-15-04268-f001:**
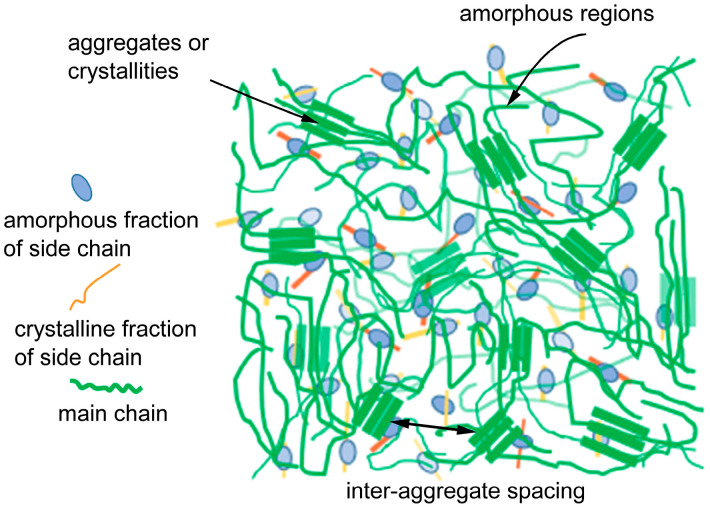
A schematic diagram of the carrier transport process in conjugated polymers is influenced by the hierarchical structure at various length scales ranging from chain conformation, ordered aggregates, and chain arrangement. These structural characteristics directly impact the efficiency and effectiveness of charge transport within the conjugated polymer system.

**Figure 2 polymers-15-04268-f002:**
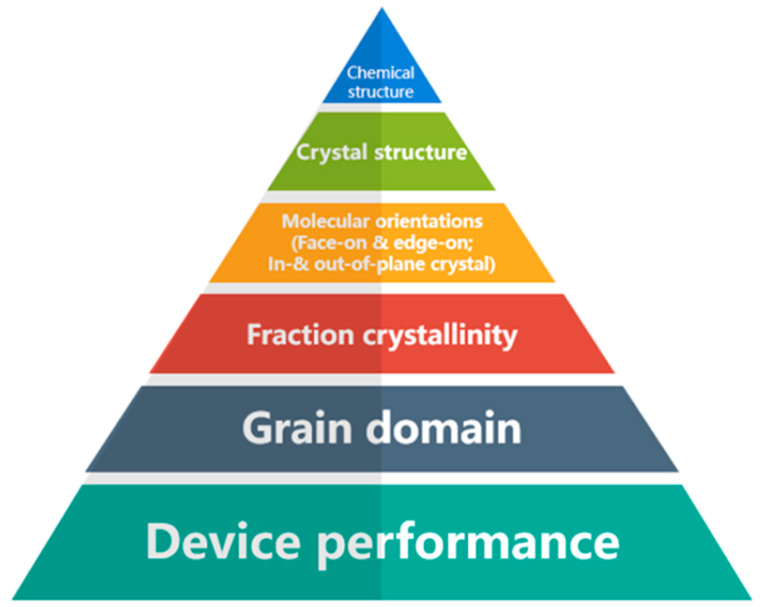
The relationship between the performance of devices and the molecular structure of the organic semiconductor (reprinted from Ref. [22] with minor modifications, with permission from the Royal Society of Chemistry).

**Figure 3 polymers-15-04268-f003:**
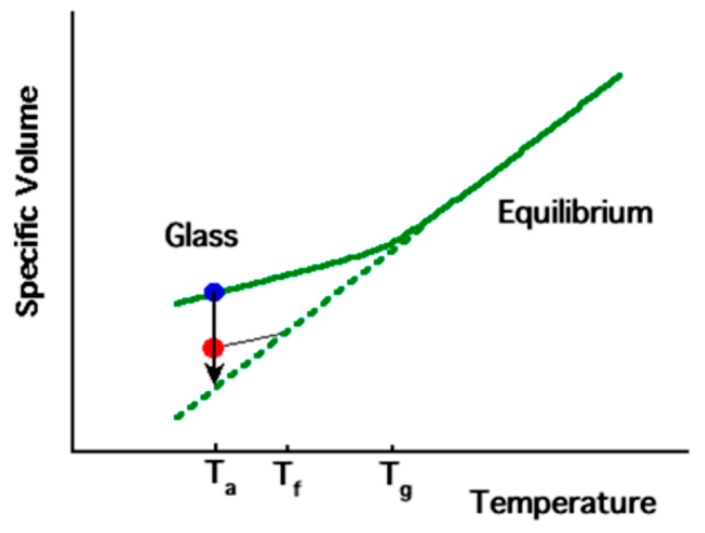
Schematic diagram of specific volume temperature dependence of typical amorphous polymer (reprinted from Ref. [37], with permission from American Chemical Society).

**Figure 4 polymers-15-04268-f004:**
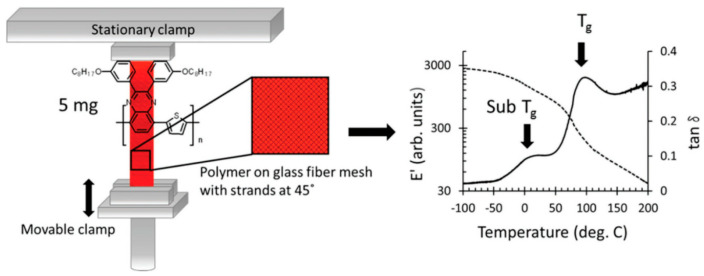
Schematic of modified DMA instrument set−up for measuring samples (reprinted from Ref. [50], with permission from American Chemical Society).

**Figure 5 polymers-15-04268-f005:**
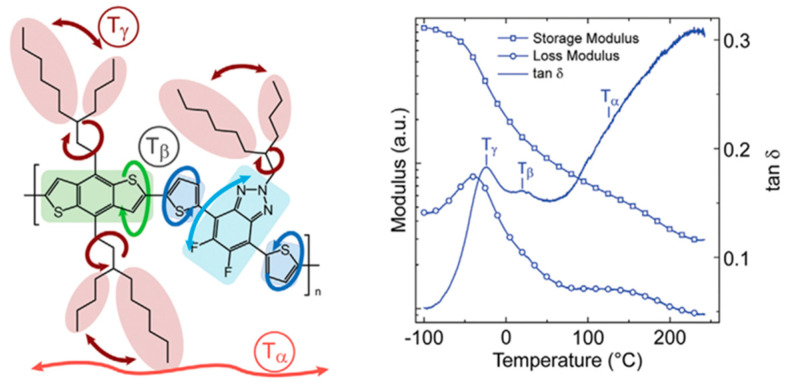
Illustration showcasing the molecular motions in PBnDT−FTAZ along with the DMA scan of PBnDT−FTAZ conducted at a frequency of 1 Hz and a temperature ramp of 3 °C/min (reprinted from Ref. [96], with permission from John Wiley and Sons).

**Figure 6 polymers-15-04268-f006:**
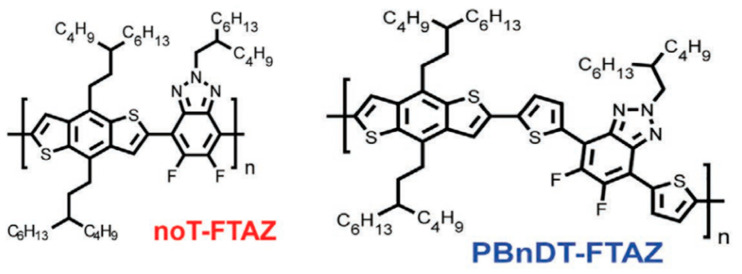
Molecular structures of the noT−FTAZ and PBnDT−FTAZ (reprinted from Ref. [96], with permission from John Wiley and Sons).

**Figure 7 polymers-15-04268-f007:**
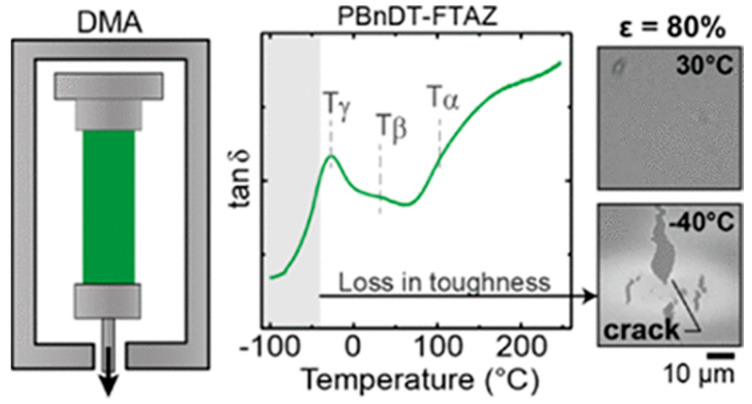
Tan *δ* curves of PBnDT−FTAZ obtained from DMA temperature sweep and the images of PBnDT–FTAZ at 30 °C and −40 °C (reprinted from Ref. [98], with permission from the American Chemical Society).

**Figure 8 polymers-15-04268-f008:**
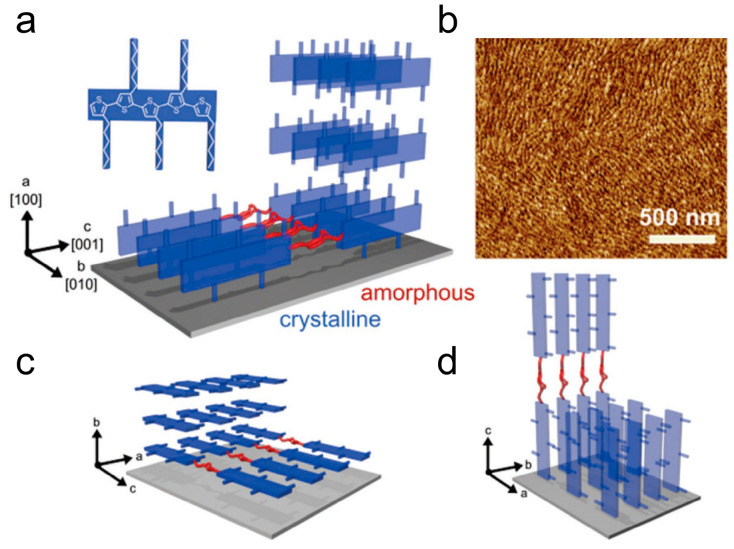
Three different π-stack textures depict the orientation of P3HT: (**a**) edge-on, (**c**) face-on, and (**d**) flat-on. (**b**) AFM phase image of P3HT spherulite grown via solvent vapor annealing (reprinted from Ref. [108], with permission from John Wiley and Sons).

**Figure 9 polymers-15-04268-f009:**
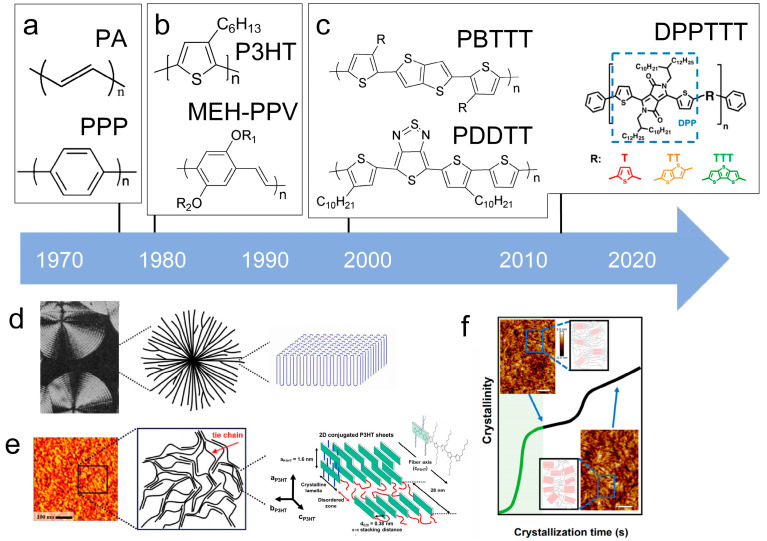
The microstructure of polymer crystallites, as well as the molecular structure of conjugated polymers. (**a**) Polyacetylene (PA) and polyparaphenylene (PPP) (reprinted from Ref. [116], with permission from the Royal Society of Chemistry), (**b**) P3HT and poly[2-methoxy-5-(2-ethylhexyloxy)-1,4-phenylenevinylene] (MEH-PPV), and (**c**) poly(2,5-bis(3-alkylthiophen-2-yl)thieno[3,2-b]thiophene) (PBTTT), poly (5,7-bis(4-decanyl-2-thienyl)thieno(3,4-b)diathiazole-thiophene-2,5) (PDDTT) and thieno[3,2-b]thiophene-diketopyrrolopyrrole (DPPTTT). (**d**) Polarized optical microscopy (POM) image of polyethylene spherulite (**left**) and schematic of spherulite structure (**middle**) composed of lamellae (**right**). (**left**) (reprinted from Ref. [117], with permission from John Wiley and Sons). (**e**) Displayed above is an atomic force microscopy (AFM) image capturing a P3HT film (**left**). A schematic representation of the molecular structure is provided, demonstrating the presence of locally ordered regions that are connected by tie chains (**middle**). Moreover, a schematic showcasing the crystal structure of P3HT is illustrated ((**left**) and (**middle**)) (reprinted from Ref. [118], with permission from the American Chemical Society), (**right**) (reprinted from Ref. [119], with permission from John Wiley and Sons). (**f**) The morphologies of the DPPTTT conjugated polymer through AFM, both after quenching from the melt and following an extended period of annealing (reprinted from Ref. [67], with permission from the American Chemical Society).

**Figure 10 polymers-15-04268-f010:**
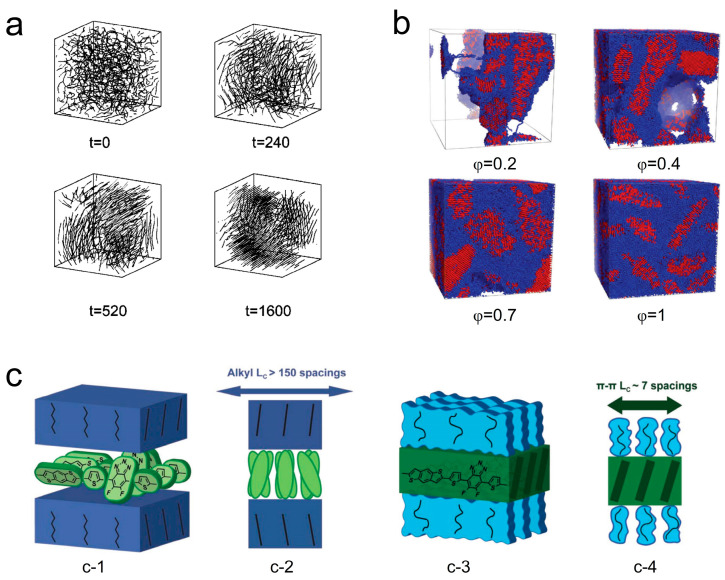
Illustrations of how rigidity, semiflexible chains, and the side chain effect can influence the crystallization include (**a**) a simulation of the crystallization process of semirigid chains (reprinted from Ref. [133], with permission from the American Physical Society) and (**b**) crystallization of semiflexible polymers with different volume fractions in melts and solutions (reprinted from Ref. [134], with permission from the Royal Society of Chemistry). (**c**) The competition mechanism for orderly arrangement between sidechains and main chains of the conjugated polymer PBnDT−FTAZ. (c-1 (front view), c-2 (side view)) Twisted main chain with disordered side chain and (c-3 (front view), c-4 (side view)) ordered main chain with disordered side chains (reprinted from Ref. [136], with permission from John Wiley and Sons).

**Figure 11 polymers-15-04268-f011:**
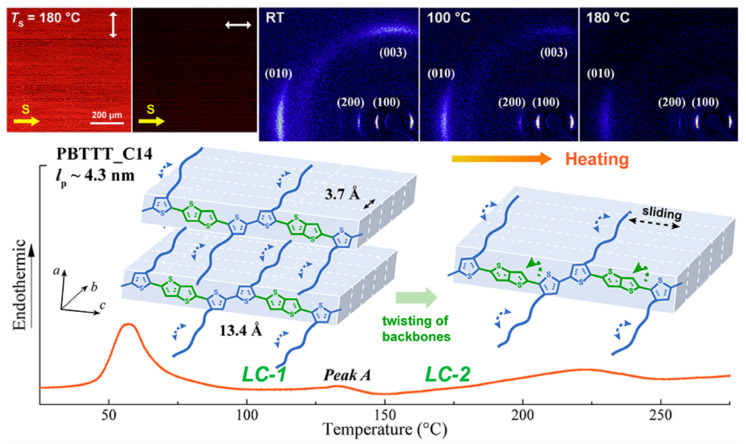
The illustration of the aggregate states’ evolution of PBTTT-C14 at different annealing processes (reprinted from Ref. [141], with permission from the American Chemical Society).

**Table 1 polymers-15-04268-t001:** *T_g_* of the conjugated polymers. [13,45,65,77,78,79,80,81,82,83].

Polymers	Chemical Structure	*T_g_* [°C]	Sidechain *T_g_* [°C]	Test Method	*M_n_* [kg/mol]	Reference
PT	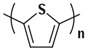	120		DMA		[77]
P3BT	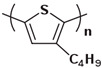	45		DMA	17	[45]
		60		DSC		[78]
P3HT	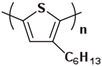	12	−87	DMA	10	[45]
		16.3		DSC		[79]
P3OT	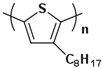	−13	−65	DMA	21	[45]
P3DT	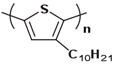	−25	−59	DMA		[45]
P3DDT	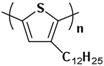	−18	−49	DMA		[45]
P3(4MP)T	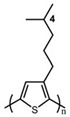	36		DSC	20.9	[79]
P3(3MP)T	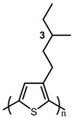	41.4		DSC	17.3	[79]
P3(2MP)T	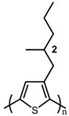	30		DSC	20.5	[80]
P3(2EB)T	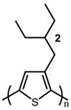	43.8		DSC	14.1	[80]
P(DPPT)	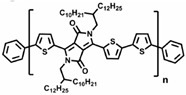	−3.96	−54.3	DMA	47	[81]
		*T_g_*_,r_: 118*T_g_*_,m_:23	−43	AC chip	47	[65]
P(DPPTT)	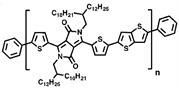	2.76	−53	DMA	51	[81]
		*T_g_*_,r_: 115*T_g_*_,m_:27	−43	AC chip	51	[65]
P(DPPTTT)	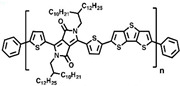	4.11	−52.5	DMA	26	[81]
		*T_g_*_,r_: 113*T_g_*_,m_:29	−43	AC chip	44	[65]
PIDTTPD	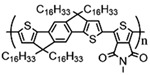	−0.6		DSC	14	[82]
PIDTBTD	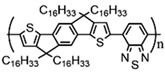	17.6		DSC	15	[82]
PIDTBPD	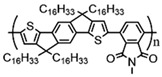	−8.1		DSC	15	[82]
IDT-BT	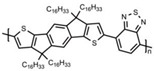	56	−20	DMA	92	[13]
NDT-BT	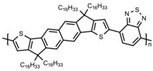	101	−20	DMA	70	[13]
TBIDT-BT	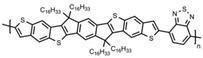	91	−20	DMA	62	[13]
TIF-BT	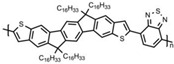	141	−20	DMA	57	[13]
poly(AZ-NDI)- 3	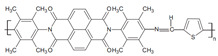	124		DSC		[83]
poly(AZ-NDI)- 4	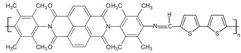	123		DSC		[83]

## Data Availability

No new data were created or analyzed in this study. Data sharing is not applicable to this article.

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
