# Peer review of "Structure–Property Relationship, Glass Transition, and Crystallization Behaviors of Conjugated Polymers"

_polymers, 2023, doi:10.3390/polym15214268_

Round 1
Reviewer 1 Report
Comments and Suggestions for Authors
This review summarizes the basic mechanism in charge transport, basis of glass transition and measurement of glass transition temperature, and relationship between the structure and mechanical properties of conjugated polymers. In addition, the crystallization behavior of conjugated polymers is also discussed. This manuscript provides a useful information for the design of new conjugated polymers in flexible electronics. However, the following issues need to be addressed before it can be accepted for publication:
1. In chapter 2, the authors discussed the multi-scale aggregated structure and the charge transport of conjugated polymers. However, the influence of molecular orientation (edge-on or face-on) on charge transport is not mentioned in this manuscript. I think the molecular orientation of conjugated polymers strongly affects their charge transport property. Therefore, the authors should provide the reasonable discussion about the relationship between the charge transport and molecular orientation of conjugated polymers.
2. In chapter 3, the discussion about the effect of chemical structure (including length and shape of side chains, and backbone structure) on glass transition temperature (Tg) and mechanical stability is insufficient. In cited references, Tgs of several D-A polymers such as indacenodithiophene-benzothiadiazole-based polymers, tetrathienoacene-based polymers, diketopyrrolopyrrole-based polymers, and so on) are reported. The authors should provide chemical structures and Tg of conjugated polymers to understand the effect of chemical structure of polymers on Tg and mechanical stability.
3. In conclusion part, the authors designed several new conjugated polymers. Why did the authors choose these polymers?
4. In Figure 7d, “flat-on” is correct? I think this type orientation is called “end-on”.
5. In page 9, the name of polymer PBnDT-FTAZ is not correct. This polymer has benzodithiophene and benzotriazole units. “non-thienyl-substituted benzothiadiazole (no-T-FTAZ)” should be “non-thienyl-substituted benzotriazole (no-T-FTAZ)”.
6. In reference part, some page numbers and author names are missing. Please carefully check it.
Reviewer 2 Report
Comments and Suggestions for Authors
The authors present an interesting review focused on the glass transition and crystallization behaviors of polythiophene-based conjugated polymers and their structure-property correlations.
The review is well-written, well-organized and it is definitely an excellent contribution to the field as it covers aspects that are usually hard to find in the literature, especially in reviews.
In my opinion, the review can be accepted for publication after addressing the following minor queries:
Q1) Line 174
It is well-written in the text but the Tg in the equation must be substituted for Tg,infinite.
Q2) Line 195
Note that these are not glass transition temperature (Tg) values. Instead, these values referred to the relaxation processes values usually called the "polyethylene-like glass transition" observed at low temperatures. It would be more appropriate to use the same nomenclature (alpha-PE) as in Ref 30 rather than Tgs. Please correct.
Q3) Line 326
Please note that according to Ref 30, the values for Tg (in Celsius degrees) obtained from DSC experiments are: 34 (P3HT), 4 (P3HT) and -22 (P3OT). Authors can use ranges to inform these values or simply clarify that these values can be slightly different.
